# An Integrative ATAC-Seq and RNA-Seq Analysis of the Endometrial Tissues of Meishan and Duroc Pigs

**DOI:** 10.3390/ijms241914812

**Published:** 2023-09-30

**Authors:** Han Zhang, Zhexi Liu, Ji Wang, Tong Zeng, Xiaohua Ai, Keliang Wu

**Affiliations:** National Engineering Laboratory for Animal Breeding, Department of Animal Genetics and Breeding, College of Animal Science and Technology, China Agricultural University, Beijing 100193, China; hanzhang@cau.edu.cn (H.Z.); bs20193040372@cau.edu.cn (Z.L.); ji.wang@cau.edu.cn (J.W.); woshizttttt@163.com (T.Z.); aliceaxh@163.com (X.A.)

**Keywords:** ATAC-seq, RNA-seq, endometrium tissue, litter size, Meishan pigs

## Abstract

Meishan pigs are a well-known indigenous pig breed in China characterized by a high fertility. Notably, the number of endometrial grands is significantly higher in Meishan pigs than Duroc pigs. The characteristics of the endometrial tissue are related to litter size. Therefore, we used the assay for transposase-accessible chromatin with sequencing (ATAC-seq) and RNA-sequencing (RNA-seq) to analyze the mechanisms underlying the differences in fecundity between the breeds. We detected the key transcription factors, including Double homeobox (Dux), Ladybird-like homeobox gene 2 (LBX2), and LIM homeobox 8 (Lhx8), with potentially pivotal roles in the regulation of the genes related to endometrial development. We identified the differentially expressed genes between the breeds, including *SOX17*, *ANXA4*, *DLX3*, *DMRT1*, *FLNB*, *IRF6*, *CBFA2T2*, *TFCP2L1*, *EFNA5*, *SLIT2*, and *CYFIP2*, with roles in epithelial cell differentiation, fertility, and ovulation. Interestingly, *ANXA4*, *CBFA2T2*, and *TFCP2L1*, which were upregulated in the Meishan pigs in the RNA-seq analysis, were identified again by the integration of the ATAC-seq and RNA-seq data. Moreover, we identified genes in the cancer or immune pathways, FoxO signaling, Wnt signaling, and phospholipase D signaling pathways. These ATAC-seq and RNA-seq analyses revealed the accessible chromatin and potential mechanisms underlying the differences in the endometrial tissues between the two types of pigs.

## 1. Introduction

Pigs (*Sus scrofa*) are the most effective mammals at converting feed into protein, and their role in animal husbandry is becoming increasingly crucial [1]. The Meishan pig breed, originating from the Taihu Lake region, is one of the most well-known native pig breeds in China and is distinguished by its high fertility, producing from three to five more piglets per litter than US or European pig varieties [2]. Meishan pigs exhibit earlier puberty, larger litter sizes, a reduced time of estrus after weaning, higher ovulation rates, and a higher density of blood vessels in the placenta compared to Duroc pigs [3,4], and the litters of Meishan pigs average 1.69 more pigs than that of Duroc pigs [5]. These economic traits are intricately linked to their uterine capacity, ovulation rate, endometrial gland development, and embryo attachment during pregnancy [6]. Meishan pigs have been introduced into various countries, such as the USA, serving as an exceptional model for hybridization and reproductive improvement [7]. The effective conservation of Meishan pigs can not only preserve their excellent traits, but also provide valuable resources for improving the reproductive performance of introduced pig breeds and breeding new breeds. Therefore, this study used the Meishan pig breed to better understand the genetic mechanisms underlying their high fecundity. 

Crossbreeding studies have shown that the characteristics of Taihu sows are determined by strong maternal effects, and maternal genes are primarily responsible for their large litter sizes [8]. This maternal effect contributes to the prolificacy of Meishan pigs, including their hormone levels and the development of their endometrial glands [9]. Endometrial glands, present in the uteri of all mammals, produce or transport substances necessary for embryo survival and conceptus growth within the uterine lumen [10]. Ovine uterine gland-knockout sheep exhibit pregnancy loss, conclusively demonstrating the need for these glands and the compounds they produce for fertility [10]. Secretory products from the endometrial glands are essential for uterine receptivity and conceptus implantation [11]. Endometrial gland development occurs shortly after birth, and the level of retinol-binding protein increases with endometrial gland development. According to previous studies, the uterine secretions in Meishan pigs peak during the early stages of pregnancy or after the maternal detection of pregnancy, which may explain why they are highly prolific [9]. On gestational day (GD) 12, the growth and development of pig embryos are dependent on histotrophs released by the endometrium. A study showed that, on GD12, Taihu pigs lost fewer embryos than the corresponding estimates for Western pigs [8]. Another study revealed that adult uterine function may be affected by the biochemical processes associated with fetal endometrial gland development [12]. Uterine receptivity and embryo implantation depend on the production of leukemia inhibitory factor and calcitonin, both of which are solely generated by the uterine glands [13]. The endometrial tissue of Meishan pigs can serve as a good model for understanding the molecular mechanisms underlying variations in litter size, indicating that the endometrial gland is an essential factor influencing this litter size. Therefore, comparative analyses of Meishan and Duroc pigs could guide the development of strategies for enhancing prolificacy.

In eukaryotic cells, nuclear DNA and proteins such as histones combine to form chromatin, which then undergoes intricate and orderly folding to form chromosomes [14]. The assay of transposase-accessible chromatin sequencing (ATAC-seq), in which transposase Tn5 is used to break open DNA regions and high-throughput sequencing is applied to investigate chromatin accessibility [15], can rapidly and sensitively detect the chromatin accessibility in an entire genome, reflect how chromatin structural changes affect gene expression [16], and aid in elucidating regulatory mechanisms using RNA-sequencing (RNA-seq) [17]. The primary transcription factors (TFs) affected by perturbations can be identified using the genome-wide profile of TF recognition motifs in relation to regions of open chromatin [18]. Chromatin accessibility represents the ability of chromatin DNA to bind to other molecules, such as transcription factors and motifs, and is closely related to gene expression levels [19]. Additionally, RNA-seq can identify differentially expressed genes (DEGs) and, to a certain extent, represent the level of gene expression in tissues at particular time points [20]. Therefore, genes with chromatin accessibility in their promoters are likely to be differentially expressed at the mRNA level and regulated by TFs [21]. Some recent studies have used ATAC-seq and RNA-seq to analyze chromatin openness characteristics to study the intramuscular fat content [22] and the longissimus or semitendinosus muscle [23,24] in pigs. However, the relationship between chromatin accessibility and gene transcription in the endometria of Meishan and Duroc pigs is poorly understood.

In this study, we compared the chromatin accessibility between Meishan and Duroc pig endometria. To identify the TFs that control the development of endometrial tissue, we used ATAC-seq and RNA-seq to detect the open chromatin areas and specific genes related to a large litter size in the two pig breeds. We integrated the ATAC-seq RNA-seq results and identified several putative core genes and TFs that may influence endometrial development during proliferation and differentiation. Several pathways controlling this process were identified. These results offer an invaluable resource for further studies into the factors that influence the distinctions between the two pig breeds.

## 2. Results

### 2.1. Phenotype of Meishan and Duroc Pig Endometrial Tissues

The litter size of Meishan pigs is higher than that of Duroc pigs, in part because their endometria are more widely distributed and numerous (Figure 1a–c). Forkhead box A2 (FOXA2), an endometrium-associated antigenic marker, is frequently used to evaluate the endometrial development in mammalian uteri [25]. The *FOXA2* gene expression levels, as determined via RT-qPCR, were higher in the Meishan group than the Duroc group (Figure 1d).

### 2.2. Quality Assessment of ATAC-Seq Data for Meishan and Duroc Pig Endometrial Tissues

To reveal the mechanisms underlying the observed differences in the endometrial tissues, ATAC-seq was processed to detect the landscape of genomic chromatin accessibility. The clean data for all the samples were used to map the genome (*Sus scrofa* 11.1) (Appendix A). The mapability was above 95%, and over 93% of clean reads, on average, were mapped in proper pairs (Appendix A), which is a moderately high ratio. We evaluated the libraries’ quality based on the lengths of the inserts and expected distributions of the peak signals (Figure 2a). Numerous earlier ATAC seq data analyses have revealed similar findings with regard to the distribution of fragment lengths. Most accessible areas were located within 2 kb of the transcription start site (TSS), suggesting that open regions of chromatin are crucial for the control of transcription (Figure 2b). The high standard of the ATAC-seq data was also proved by the mapped read distributions, spanning gene bodies and peaks. 

Pearson’s correlation coefficients revealed the difference between the two pig breeds based on the read signals, further demonstrating the accuracy of our ATAC-seq data. The nearer the correlation ratio was to 1, the more closely the two samples’ expression patterns matched up (Figure 2c). A principal component analysis (PCA) revealed similarities between the biological replicates and differences, with clustering by group according to the amount of data variability, that could be explained in PC1 (97.7%) and PC2 (1.9%). The PCA results indicated clear variations in the transcriptome profiles between the two breeds. (Figure 2d). These imply that the sequencing data had an excellent quality.

### 2.3. Chromatin Accessibility in Endometrial Tissues of Meishan and Duroc Pigs

We identified 99,908 peaks specific to the Meishan pigs, 39,812 peaks specific to the Duroc pigs, and 36,154 common peaks (Appendix A). The MA plot displays the correlation between all the peaks’ open intensity and fold changes; the chromatin open region map at the chromosome level in the whole genome shows the difference in the openness of all the peaks (Figure 3a). Both the signal strength and distribution of the peaks along the chromosomes of the pig genome were consistent between the two groups (Figure 3b). To annotate all the peaks, functional sections were separated into exons, promoters, introns, intergenic regions, 5′ untranslated regions (UTRs), and 3′ UTRs. The majority of the peaks were mapped to these regions in the promoter, intron, and intergenic regions (Figure 3c). ATAC-seq signals are usually enriched in open chromatin regions and positively correlated with gene transcription. A heatmap reflected the enrichment of the reads in the regions 5 kb upstream and downstream of the TSS and termination end sites (TESs) of all the genes in the genome. The Meishan pigs had more active TFs than those of the Duroc pigs, with more binding sites and more active sites related to biological activities (Figure 3d).

### 2.4. Analyses of Genes and Motifs in Different Peaks

To identify the open chromatin regions related to the phenotypic variation, there were 1785 different peaks for the Duroc pigs and 6787 different peaks for the Meishan pigs (Appendix A). Further annotation revealed that the different peaks for the Duroc pigs corresponded to 228 genes and those for the Meishan pigs corresponded to 1960 genes, resulting in a total of 2188 genes. Gene Ontology (GO) and Kyoto Encyclopedia of Genes and Genomes (KEGG) pathway enrichment analyses of these genes were performed, with *p* ≤ 0.05 indicating significant enrichment. The GO enrichment analysis revealed enrichment for the regulation of metabolic processes in the biological processes category, organic cyclic compound binding or carbohydrate derivative binding in the molecular function category, and the extracellular part for the cellular component category (Figure 4a). Moreover, various KEGG pathways were significantly enriched (*p* < 0.05), in particular cancer or immune pathways and the FoxO signaling, Wnt signaling, and glutamatergic synapse pathways (Figure 4b). The motifs on the different peaks for the Meishan pigs were examined by the Multiple Expectation maximizations for Motif Elicitation (MEME) suite and the TFs were expected by the TOMTOM motif database. Finally, 1028 TFs (*p* < 0.05) were predicted in the Meishan pigs. In addition, several key TFs related to embryonic development and reproduction were significantly enriched at the top ATAC-seq peak in the MS pigs, suggesting that endometrial development is related to the binding of TFs to open chromatin regions. The TFs Dux, LBX2, and Lhx8 are key regulators of the expressions of many genes related to endometrial development, as shown in Table 1.

### 2.5. RNA-Seq Data from Endometrial Tissues of Meishan and Duroc Pigs

The total RNA extracted from the endometrial samples yielded transcript data free from genomic, protein, or impurity contamination, with no color abnormalities. The RNA-seq results are shown in Appendix A. Over 92% of clean reads were successfully mapped to the pig genome, and all the mapped reads were distributed in the genome, as shown in Figure 5a. The proportion of reads aligned to exonic regions was the highest in all samples, which indicated that the data were reliable.

To identify genes associated with the observed phenotypic differences, a differential expression analysis was conducted according to the criteria of |log2(fold change)| ≥ 1 and *p* < 0.05. There were 2372 DEGs between the Meishan and Duroc groups, 1158 of which were upregulated and 1214 were downregulated in Meishan pigs (Figure 5b, Appendix A). Heatmaps of these DEGs show that the expression profiles remained similar within groups but different between groups. (Figure 5c). Assessments of GO and KEGG pathway enrichment were used to analyze upregulated and downregulated mRNAs. In the biological process category, the DEGs were enriched in several cellular and biological processes; in the cellular components category, the DEGs were mainly detected in the membrane; and in molecular function category, the DEGs were primarily concentrated in binding and catalytic activity (Figure 5d). To explore the potential biological functions of the DEGs, a functional enrichment analysis was performed using Metascape with *p* ≤ 0.05 as the threshold. The upregulated genes in Meishan pigs were mainly involved in epithelial cell differentiation, tissue morphogenesis, and embryonic organ development (Figure 5e). Some of the DEGs involved in these pathways, such as *SOX17*, *ANXA4*, *DLX3*, *DMRT1*, *FLNB*, *IRF6*, *CBFA2T2*, *TFCP2L1*, *EFNA5*, *SLIT2*, and *CYFIP2*, were involved in epithelial cell differentiation, regulation of fertility, and ovulation. Numerous genes were engaged in cell adhesion, cytokine–cytokine receptor interactions, and neuroactive ligand receptor interactions, according to a KEGG pathway enrichment analysis. (Figure 5f).

### 2.6. Integration of ATAC-Seq and RNA-Seq Results

To further understand more about the association between the gene expression and open chromatin regions, we integrated the RNA-seq and ATAC-seq results. Based on the RNA-seq, we obtained 1158 upregulated genes and 1214 downregulated genes in the MS vs. DR comparison. Based on ATAC-seq, we obtained 1960 peaks with stronger signals and 228 differential peaks with weaker signals in the MS vs. DR comparison. In total, 392 overlapping DEGs emerged, 223 of which showed an increase in their expression and 169 showed a decrease in MS vs. DR (Figure 6a, Table 2 and Appendix A). These DEGs were enriched in the pathways related to organic homeostasis and maternal placental development (Figure 6b). Interestingly, *ANXA4*, *CBFA2T2*, and *TFCP2L1*, which were upregulated in MS using RNA-seq, were identified again using ATAC-seq. A KEGG analysis showed that some enriched pathways, such as the phospholipase D and sphingolipid signaling pathways, were related to cell proliferation and the regulation of immune and endothelial differentiation (Figure 6c). 

### 2.7. Validation of RNA-Seq Results Using qRT-PCR

To validate the accuracy of the RNA-seq results, nine randomly selected DEGs (*SOX17*, *WNT6*, *ADCY6*, *RIMKLB*, *NOP2*, *DMRTI*, *BRINP2*, *DLX3*, and *ANXA4*) were analyzed using qRT-PCR. The expression patterns of these genes were consistent with the RNA-seq data (Figure 7, Appendix A).

## 3. Discussion

Among the various reproductive traits, litter size holds paramount importance as an economic trait and plays a crucial role in the swine industry [26]. Litter size is a polygenetic trait determined by multiple gene regulatory pathways [27]. However, achieving genetic progress and stability remains a challenge, as does maintaining a balance between an increased litter size and decreased mortality rates in piglets [28]. The quantity and distribution of the pig endometrium and uterine glands are important factors that influence the implantation of fertilized eggs and litter size [29]. Studies have demonstrated that the leukemia inhibitory factor secreted by the uterine glands is essential for embryo implantation [30]. The number and position of embryos have been found to be consistent with their distribution in the uterine glands [31]. Our results indicated that the number of endometrial glands was greater in the Meishan pigs than the Duroc pigs, consistent with the results of many other studies [32]. The integration of the ATAC-seq and RNA-seq analyses could effectively identify the key factors associated with litter size, as well as the target genes of the relevant TFs in the two pig breeds, providing insights into the molecular basis of this trait. 

In this study, we detected differences in the chromatin accessibility and gene expression within the endometria of Meishan and Duroc pigs to identify the key TFs involved in the development of the endometrium and uterine glands. Initially, we identified 1785 downregulated and 6787 upregulated accessible chromatin in the Meishan group compared to the DR group; these peaks corresponded to 228 and 1960 genes, respectively. A GO analysis showed enrichment in the regulation of metabolic processes, along with organic cyclic compound binding or carbohydrate derivative binding and extracellular components. KEGG pathway analyses showed that the genes were involved in cancer or immune pathways, as well as the FoxO, Wnt, and glutamatergic synapse pathways. Interestingly, many studies have reported a correlation of the FoxO signaling pathway with fat deposition in pigs [33,34]. Zhao et al. detected enrichment for the FoxO signaling pathway in Meishan pigs via whole-genome resequencing [35]. Additionally, the endometrial tissue of Yorkshire pigs was found to express DEGs related to the FoxO signaling pathway [36]. Our results were consistent with these previous findings. The remarkable conservation of the Wnt signaling pathway in multicellular life forms reflects its indispensable role in development, and the Wnt/beta-catenin signaling pathway plays an essential role in the regulation of cell fate and polarity during the embryonic development of many animal species [37]. 

A motif is a specific base sequence with a high affinity for a TF. In this study, we analyzed the motifs on the peaks of the Meishan pigs and predicted the TFs using a motif database. Consequently, 1028 TFs (*p* < 0.05) were predicted in the Meishan pigs. In addition, we found that several key TFs related to embryonic development and reproduction were significantly enriched in these ATAC-seq peaks of the Meishan pigs, suggesting that endometrial development is related to the binding of TFs to open chromatin regions. The TFs Dux, LBX2, and Lhx8 are key regulators of the expressions of numerous genes related to endometrial development. Dux is a double homeodomain protein that promotes embryonic genome activation. It is transiently expressed at the two-cell stage and acts as a transcriptional activator during zygotic genome activation in embryos [38,39]. Ladybird homeobox 2 (LBX2) is highly expressed in various tumors and is functionally linked to the regulation of essential tumor-related biological processes, such as cell proliferation and apoptosis. This regulation occurs through interactions with multiple signaling molecules and pathways [40]. LBX2 is expressed not only in the male gonad, but also in the somatic cells of the female gonad; however, its expression in the ovary remains low until the onset of puberty. This is in sharp contrast to the male gonad, which shows a high Lbx2 expression throughout embryonic and postnatal testicular development [41,42]. The LIM homeobox 8 (*LHX8*) gene encodes an LIM homeodomain transcriptional regulator that is preferentially expressed in germ cells and is critical for mammalian oocytes [43]. LHX8 is the primary element that regulates the expression of bone morphogenetic protein 15 in oocytes [44]. Consequently, these TFs warrant attention, and their potential regulatory roles and epistatic modifications in endometrial and uterine gland development are of great interest.

We compared the gene expressions in the endometria between the Meishan and Duroc pigs, revealing 1854 DEGs, 1021 of which were upregulated and 833 were downregulated in the Meishan pigs. A functional enrichment analysis indicated that the upregulated genes in the Meishan pigs were mainly involved in epithelial cell differentiation, tissue morphogenesis, and embryonic organ development. The related pathways included various important genes, such as *SOX17* [45], *ANXA4* [46], *DLX3* [47], *DMRT1* [48], *FLNB* [49], *IRF6* [50], *CBFA2T2* [51], *TFCP2L1* [52], *EFNA5* [53], *SLIT2* [54], and *CYFIP2* [55], involved in epithelial cell differentiation, the regulation of fertility, and ovulation. Additionally, several genes were related to immunity. Several studies have showcased elevated immunity in the spleen and serum of Meishan pigs [56,57]. It is possible that the endometrium in early normal pregnancy is enriched in innate immune cells, particularly uterine natural killer (uNK) cells. Attachment sites exhibit around a three-fold enrichment of uNK cells in species with epitheliochorial placentation. uNK cells are highly enriched during decidualization and promote endometrial angiogenesis. Interestingly, at healthy sites, angiogenesis is more robustly promoted by lymphocytes than by trophoblasts [58]. In conclusion, these genes and pathways warrant further attention, and their interactions with TFs and epistatic modifications in endometrial development should be a focus of future research.

Significantly, this study exclusively examined genes that were positively regulated in the endometrium and uterine glands. However, more information can be mined for further analysis and verification, including genes related to other tissues, functional data for other modular genes, and joint analyses with other epistatic data.

## 4. Materials and Methods

### 4.1. Sample Collection

Meishan (*n* = 3) and Duroc (*n* = 3) swine aged 210 days during the estrus stage were divided into two groups with extremely high and low fecundity (MS1, MS2, MS3, DR1, DR2, and DR3). Endometrial samples were collected and placed in liquid nitrogen for the ATAC-seq and RNA-seq analyses. All the animals were provided by the Kunshan Meishan Pig Breeding Farm and Beijing Shunxin Agricultural Duroc Breeding Farm. The animal collection and experiments were approved by the Institutional Animal Care and Use Committee of China Agricultural University (no. AW010802202-1-1). 

### 4.2. Histological Analysis

To determine the number of endometrial cells, fresh tissues were washed with phosphate-buffered saline, immobilized in a 4% paraformaldehyde solution for 24 h, and dehydrated by an alcohol gradient, with transparency and dipping in wax. The dipped uterine tissues were quickly placed in the embedding base containing partially dissolved wax and left to cool. A solidified paraffin block was used to prepare 5 µm serial cross-wise sections, followed by staining with hematoxylin and eosin for 10 min. Histological examinations were performed using a light microscope (Leica, Wetzlar, Germany). Each pig was sampled for three tissue sections, and five visions were randomly selected for each section at 40× magnification. The images were taken under an upright microscope.

### 4.3. ATAC-Seq

Libraries for the ATAC-seq were generated using six samples. The nuclei were extracted and resuspended in the Tn5 transposase reaction mixture. The transposition reaction was incubated at 37 °C for 30 min, and then equimolar adapter 1 and 2 were introduced. The library was then amplified using PCR, followed by purification with AMpure beads. The quality was tested using Qubit. Clustering of the index-coded samples was carried out on a cBot cluster generation system using HiSeq PE Cluster Kit v4-cBot-HS (Illumina). The libraries were sequenced on an Illumina platform after the cluster generation and 150 bp paired-end reads were produced [59]. The original Illumina high-throughput sequencing image data files were transformed into original sequences after base group recognition using the bcl2fastq software, and were stored in the FASTQ file format. High-quality clean reads were acquired with Trimmomatic (v0.36), and the sequences were compared with the reference genome (*Sus scrofa* 11.1) using Bowtie2. For peak detection, MACS2 (v2.1.1) was run to obtain an overview of the open chromatin regions in the entire genome of each sample. Then, bedtools (v2.17.0) was used to merge the peaks to identify differences in the accessibility. Each sample’s various peaks corresponded to genes that were recognized and annotated. There were three biological replicates used. The TOMTOM motif database-scanning algorithm was used to predict the TFs after the motif analysis with the MEME suite.

### 4.4. RNA-Seq

Illumina high-throughput sequencing served to sequence the total RNA from the endometrial samples. The RNA sample preparation process used a total of amount of 3 µg of RNA per sample as the initial substance. Sequencing libraries were created using the NEBNext^®^ Ultra™ RNA Library Prep Kit for Illumina^®^ (NEB, Ipswich, MA, USA), in accordance with the manufacturer’s instructions. On the Agilent Bioanalyzer 2100 equipment, the library quality was measured. The clustering and sequencing were consistent with the methods used for the ATAC-seq [59]. Original data (raw reads) were accessed and transferred to sequencing data via base calling to obtain clean reads. Clean, high-quality data served as the foundation for all the downstream studies. Then, Hisat2 v2.0.5 was used to compare the sequences with the reference genome (*Sus scrofa* 11.1) FeatureCounts (v1.5.0) for counting reads linked to each gene. The length of each gene and the amount of reads that were mapped to it were then used to determine the FPKM value for each gene. ‘DESeq2’ (1.20.0) in R was used to perform a differential expression analysis. The false discovery rate was obtained by adjusting the *p*-value, and the following standards were used to determine which DEGs were significant: *p*adj ≤ 0.05 and |log2.Fold_change| ≥ 1. To explore the unique biological functions and important pathways between the Meishan and Duroc pigs, we evaluated the DEGs using Gene Ontology (GO) and Kyoto Encyclopedia of Genes Genomes (KEGG) analyses and the Metascape website (http://metascape.org (accessed on 30 May 2022)).

### 4.5. Integration of ATAC-Seq and RNA-Seq

The ATAC-seq results were combined with the expression results from the RNA-seq analysis to compare the DEGs in the open chromatin regions and key TFs. The expression profiles of the peak-related DEGs in the ATAC-seq results were compared with those of the DEGs in the RNA-seq results. The upregulated and downregulated DEGs in the RNA-seq analysis were compared with the genes associated with the differential peaks in the ATAC-seq. Genes with the same expression profiles were used for the GO and KEGG pathway enrichment analyses.

### 4.6. Gene Expression and Statistical Analyses

To validate the relative expression patterns obtained from the RNA-seq, RT-qPCR was performed with samples from six Meishan and Duroc pigs. The primer information and annealing temperatures are listed in Appendix A. The PCR conditions and cycling parameters were as described previously [60]. The relative gene expression levels were calculated using the 2^−ΔΔCt^ method. All the data are expressed as means ± SEM. Differential expression was analyzed using a *t*-test and one-way ANOVA test, with values of *p* < 0.05 (*) or *p* < 0.01 (**) indicating significance. Three biological replicates were used.

## 5. Conclusions

Taken together, this study presents a novel resource for identifying the open chromatin regions and TFs implicated in the regulation of litter size in Meishan and Duroc pigs. Notably, we predicted that the TFs Dux, LBX2, and Lhx8 are key factors regulating the expressions of the genes related to endometrial development. RNA-seq revealed various genes, including *SOX17*, *ANXA4*, *DLX3*, *DMRT1*, *FLNB*, *IRF6*, *CBFA2T2*, *TFCP2L1*, *EFNA5*, *SLIT2*, and *CYFIP2*, with differential expressions between the two groups involved in epithelial cell differentiation, the regulation of fertility, and ovulation. Intriguingly, *ANXA4*, *CBFA2T2*, and *TFCP2L1* were upregulated in the Meishan pigs, as revealed by the RNA-seq, and were again identified by the integration of the ATAC and RNA-seq data. The results of this study can be used as a reference for future research on litter size differences between Meishan and Duroc pigs.

## Figures and Tables

**Figure 1 ijms-24-14812-f001:**
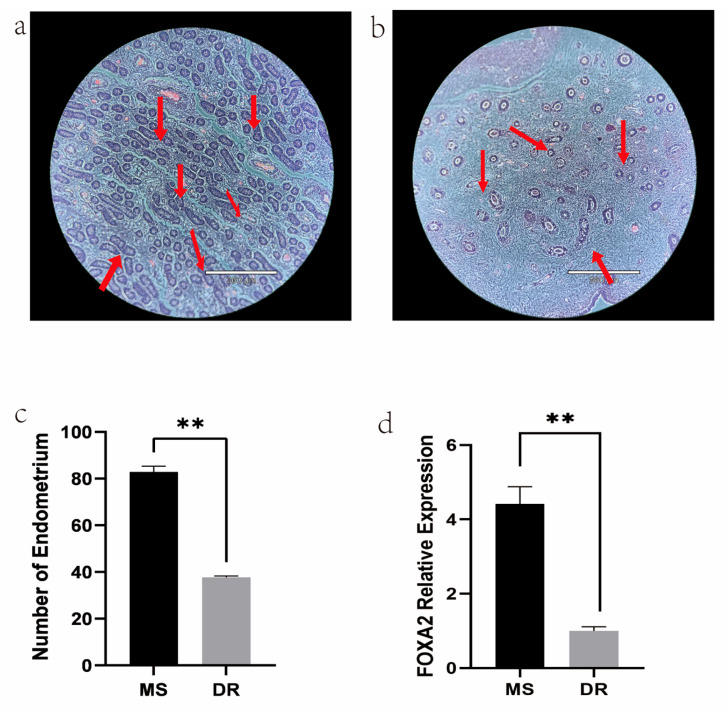
Phenotypic characteristics of the endometria. (**a**) Endometrial tissue of Meishan sections at 40× magnification, scale bar: 360 μm. (**b**) Endometrial tissue of Duroc sections at 40× magnification, scale bar: 360 μm. Endometrial glands are denoted with red arrows. (**c**) Number of endometrial glands. The data are expressed as the mean ± standard deviation (SD). ** *p* < 0.01. (**d**) Relative expression of *FOXA2*, involved in endometrial development, ** *p* < 0.01.

**Figure 2 ijms-24-14812-f002:**
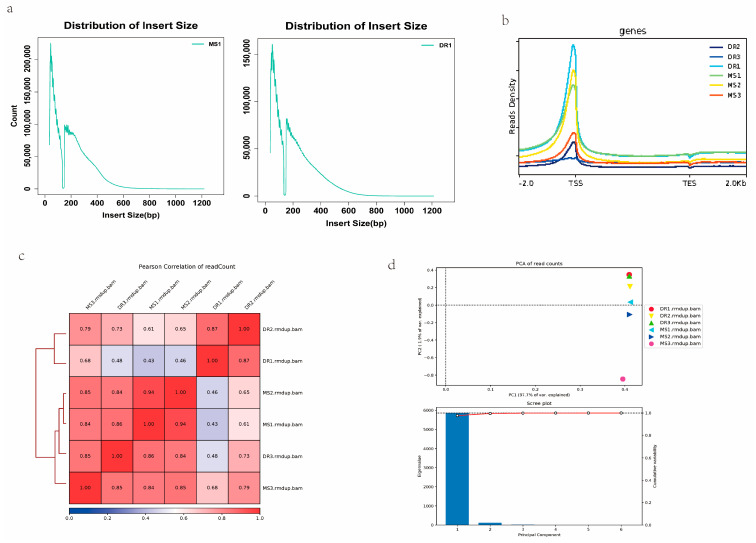
ATAC-seq quality control. (**a**) Distribution of insert sizes. (**b**) ATAC-seq signal enrichment surrounding the TSS. The read enrichment is represented by the *y*-axis, while the normalized gene or peak length is represented by the *x*-axis. The larger the value, the higher the enrichment. TSS, transcription start site; TES, transcription end site. −2.0 means for 2 kb upstream of the TSS, and 2.0 means for 2 kb downstream of the TES, respectively. (**c**) Pearson correlation coefficients displayed by heatmap scatterplot. (**d**) PCA plot.

**Figure 3 ijms-24-14812-f003:**
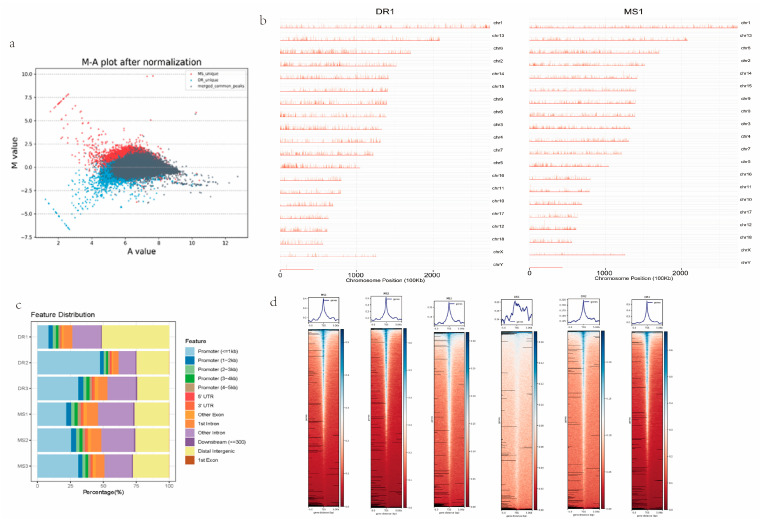
Analyses of peaks. (**a**) MA plot. (**b**) Distribution of peaks on the chromosomes. (**c**) Distribution of functional regions within different peaks. (**d**) Enrichment of ATAC-seq signals around the 5 kb upstream and downstream regions of the peak center.

**Figure 4 ijms-24-14812-f004:**
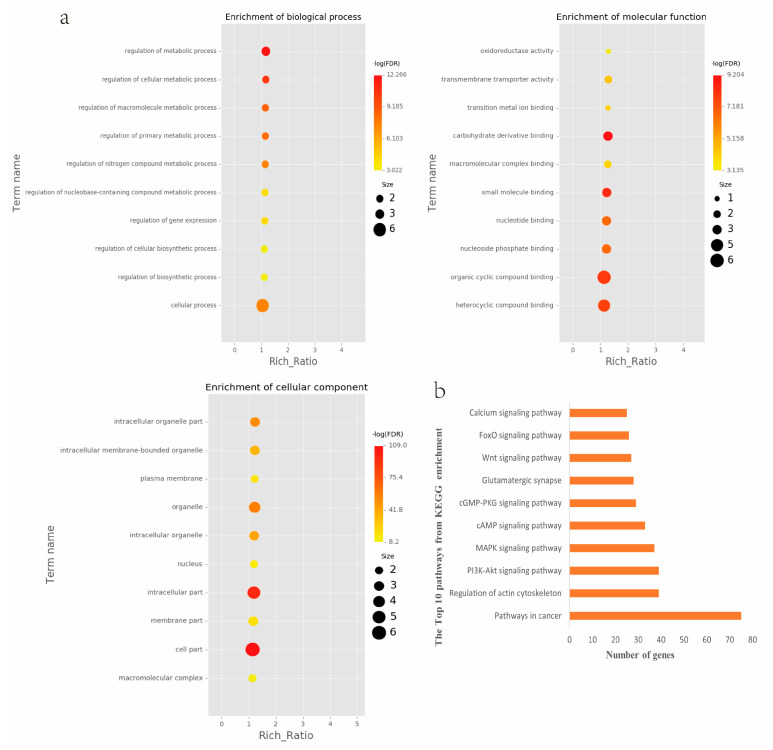
Peak differences, Examinations of relevant genes’ enrichment and predictions of transcription factors. (**a**) GO enrichment analysis of target genes related to different peaks. (**b**) KEGG pathway enrichment analysis of target genes related to different peaks.

**Figure 5 ijms-24-14812-f005:**
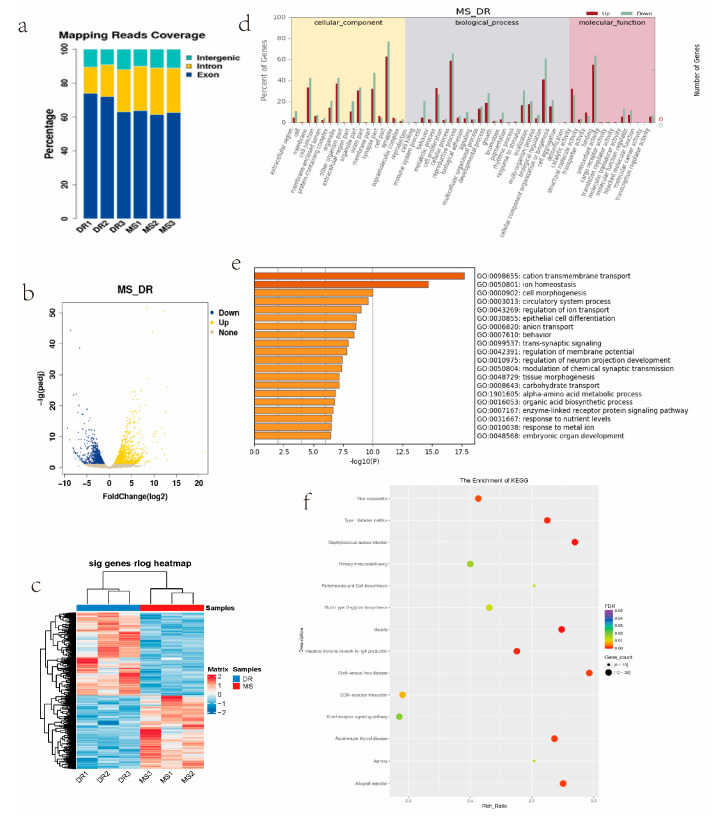
RNA–seq analysis. (**a**) Distribution of mapped reads. (**b**) Volcano plot of the transcriptome data, *p* < 0.05, |log2(fold change)| ≥ 1. (**c**) Heatmap of the genes differentially expressed between the MS and DR group. (**d**) GO enrichment analysis of differentially expressed genes. (**e**) GO enrichment analysis of upregulated genes in MS (**f**). Analysis of differentially expressed genes’ KEGG pathway enrichment.

**Figure 6 ijms-24-14812-f006:**
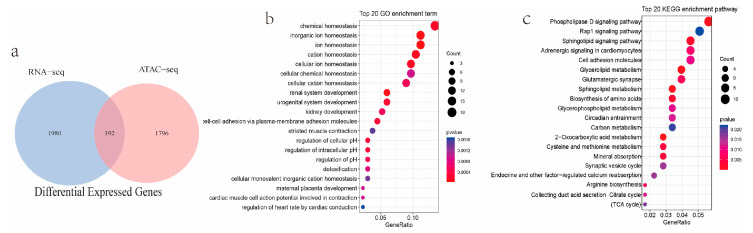
Integration of data from ATAC-seq and RNA-seq. (**a**) Converge with the genes variation in expression from ATAC-seq and RNA-seq data. (**b**) Top 20 GO enrichment analysis of differentially expressed genes. (**c**) Validation of KEGG pathway that variably expressed genes.

**Figure 7 ijms-24-14812-f007:**
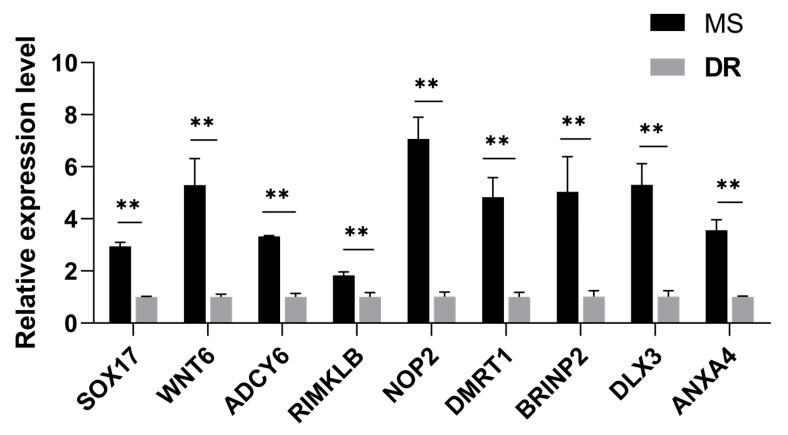
Verification of the relative expression of DEGs. The *x*-axis displays the DEGs, and the *y*-axis displays the relative degrees of expression. Data are exhibited as mean ± standard deviation (SD), *p* < 0.01, was considered statistically significant (**).

**Table 1 ijms-24-14812-t001:** Top 10 predicted binding motifs in the open chromatin region.

Number	Motifs	TF ID	E-Value
1	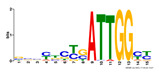	NFYA	0.000071539
2	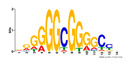	NFYB	0.081957
3	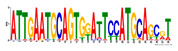	Dux	0.166414
4	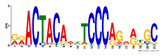	LBX2	0.934459
5	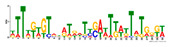	MSX1	1.61321
6	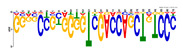	Nkx2-5(var.2)	1.62006
7	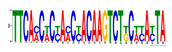	EN1	2.37482
8	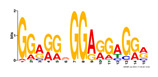	Lhx8	2.50521
9	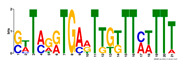	Msx3	2.93158
10	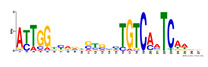	GBX2	2.93158

**Table 2 ijms-24-14812-t002:** Chromatin openness of differentially expressed genes.

	Upregulated Gene Number	Downregulated Gene Number	Total
ATAC-seq	1960	228	2188
RNA-seq	1158	1214	2372
Overlap	223	169	392

## Data Availability

The data presented in this study are available in the article or Appendix A here.

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
