# Peer review of "An Integrative ATAC-Seq and RNA-Seq Analysis of the Endometrial Tissues of Meishan and Duroc Pigs"

_ijms, 2023, doi:10.3390/ijms241914812_

Round 1

Reviewer 1 Report

Dear authors, 

I see interesting reports of endometrial tissue being tested during estrum in low and high-reproductive pigs. The molecular activity was estimated based on RNA-sea and ATAC-seq analysis. The authors used only three samples per group, but the samples clearly clustered into groups, so the number of used samples is sufficient. The manuscript is well written, but some issues should be clarified or some information or validation stage.

Introduction:

there can be added information about the differences in reproduction traits between Duroc and Meishan

in the material and methods:

Were the pigs housing and feeding in the same conditions when waiting for estrus? 

There is no information about ATAC-seq and RNA-seq library preparation and sequencing. If they were performed by service, this information should be included.

The validation stage of RNA-seq results is missing which usually is performed using qPCR.

MACS2 is the tool for peak calling, so which overlay on mACS2 or different tools were used for identifying ATAC signals showing different accessibility between analysed groups ( in your study up and down-regulated)? 

upregulated genes concern Meishan and downregulated genes were increased in Duroc; Please clarify 

In the ATAC particular, in my opinion, we should not discuss up and down-regulated ATAC peaks, but with peaks with more abundance or more accessible chromatin or less.

Author Response

Thanks  for comments and suggestions concerning our manuscript. Please see the attachment ----point-by-point response to the reviewer’s comments.

Reviewer 2 Report

Comments about the manuscript:

“Integrative ATAC-seq and RNA-seq analysis of the endometrial tissues of Meishan and Duroc pigs”

The Meishan pig is a breed native to China, with high fertility linked to the presence of a larger endometrium than in other breeds (Duroc pigs in particular). The aim of the work was to highlight and analyze the physiological phenomena involved in the different fecundities using various molecular biology methods (test of chromatin accessible to transposase with sequencing, RNA sequencing). The authors showed that different transcription factors are involved in the regulation of genes linked to the development of the endometrium.

This work provides useful elements for understanding the differences in fertility between different pig breeds. It could therefore be published after, however, a few improvements to the manuscript. Here are some remarks.

Page 3, figure 1. There is no legend for fig 1d; Put a scale bar instead of indicating magnification (the scale bar already seems to be placed in Figures 1a and b).

Page 4, figure 2d.  “Principal component analysis.”: some explanation, even brief, would be useful.

Page 4, figure 3. some explanations on the figure would be useful (I did not find these explanations in the text).

Page 10, “Histological Analysis and Immunofluorescence”. Contrary to what the title of the paragraph says, no method concerning immunofluorescence is given. Was this technique really used? It's not very clear. Please check and clarify.

“and then completely embedded in paraffin”: briefly describe the method.

“was cut crosswise”: what does "was cut crosswise " mean? Are they transverse, sagittal, longitudinal cuts. How thick were the cuts?

“Histological examinations were performed using a light microscope”: specify the microscope model (manufacturer, reference).

Supplementary material: There are figures 1, 2, 4 and 5, no figure 3. (In the text figures S1, S2, S3 and S4 are called): please, check the figures.

Author Response

(The authors gave the same response as above.)

Round 2

Reviewer 1 Report

Dear authors, I see that you carefully read my suggestions and addressed them. So, I think the manuscript can be published in its present form.

Author Response

We wish to thank the editor and reviewers for their comments and suggestions concerning our manuscript. We have revised the manuscript based on the comments and suggestions, including replacing the images with ones of sufficiently high resolution and checking that all references are complete and accurate. We have indicated major changes in red.

We feel these changes have greatly improved our manuscript and thank you for considering our revision.